# Digital-Twin-Based High-Precision Assembly of a Steel Bridge Tower

**Jiulin Li [1,*], Qingquan Li [2], Qingzhou Mao [3] and Hao Xu [1]**

1    Beijing Urban Construction Group Co., Ltd., Beijing 100088, China
2    Shenzhen University, Shenzhen 518060, China
3    College of Remote Sensing Information Engineering, Wuhan University, Wuhan 430079, China
*    Correspondence: lijiulin@bjucd.com

**Abstract:** Steel structures that benefit from having lightweight, ductility, and seismic behaviors are capable of improving the overall performance of civil engineering in environmental protection, project quality, process management, and ease of construction, making the procedure more feasible for builders. The application of steel structure techniques has been widely used in bridges, tall buildings, and complex projects around the world. Increasing demand for planning and design has led to structural projects upgrading in structural complexity and geometrical irregularity. However, steel structure projects are still limited by the principal disadvantage of susceptibility to deformation. Therefore, the challenges of the assembly and manufacturing process for steel structures are important. In this paper, to achieve full-loop tracking and control of the assembly and manufacturing process, we propose an integrated approach to undertake the aforementioned challenges via digital twin technology, which combines three modules: (1) deformation detection, (2) pose estimation and optimization, and (3) deformation correction and pose control. This proposed methodology innovatively merges gravitational deformation analysis with geometrical error analysis. Furthermore, the validity of this method's implementation is demonstrated by the New Shougang Bridge project. The results show that the assembly precision satisfies the standard of less than H/4000, nearing H/6000. Moreover, the elevation difference is less than 20 mm, which satisfies the control precision of the geometric pose. The new method that we propose in this paper provides new ideas for structural deformation control and high-precision assembly, as it realizes dynamic deformation sensing, real-time deviation analysis and manufacturing, and efficient optimization of the assembly process.

**Keywords:** steel structure; digital twin; precision control; position tracking; deviation analysis; dynamic assembly; construction technology

## 1. Introduction

Because of their characteristics of high strength, lightweight, adequate ductility, favorable processing properties, good welding performance, fast construction speed, and beneficial seismic behaviors, steel structures are popularly used in industrial buildings, long-span structures, and high-rise buildings [1–4]. Using steel structures can reduce dead weight by more than a third compared with concrete structures. Furthermore, earthquake effects can be reduced by 30~40% for bridges when the properties of steel are combined with professional bridge engineering design, materials, and equipment utilization [5]. Construction with steel allows for lower environmental damage such as the production of tiles, lime, cement, and sand excavation, and it reduces construction waste, industrial dust, and noise [5,6]. Based on institutional research, the carbon emission of conventional concrete construction is 740.6 kg per square meter, while steel construction requires about 120 kg of steel per square meter, and the carbon emission is 480 kg per square meter with a reduction of around 35% on average compared to conventional concrete construction [5]. As an example, the Hong Kong–Zhuhai–Macao bridge, which utilizes two artificial islands

in the sea and a sectional immersed tunnel between them, has a length of around 55 km, making it the longest bridge in the world [7]. Based on the requirement of a 120-year design lifetime, the whole project has a high demand for structural durability [7]. Eventually, with all factors under consideration, the bridge was decided to be a steel structure [7,8].

Steel structures in construction are sensitive to deformation [9,10]. Briefly, the current study focuses on the manufacturing process, installation process, 3D reconstruction, and reverse modeling [4,9,11]. For the assembly and manufacturing process, engineering quality control techniques, such as the thick plate welding technique and the twisted components machining technique, are still a popular topic around the world [6]. Moreover, methods involving computer science, such as the BIM technique, have been applied to construction for a while [12]. Based on this development, for example, 3D reconstruction in the data cross-validation process has been upgraded in various ways, such as the use of deep learning, computer vision, and multiple views [13,14]. An integrated model, benefiting from 3D reconstruction, structural analysis software, and sensors, can be used to construct a digital-twin-based framework to map a dynamic workflow in situ with simulations [15,16]. However, many questions still need to be answered. For instance, recent research has focused on the overall process of engineering projects, with either manufacturing or assembly. This separation has created an adjustment gap between deviations in manufacturing and variations in assembly due to the lack of mapping among digital models, production lines, and in situ methods [17]. Furthermore, informationization levels and methods of data acquisition still need perfecting and are undergoing improvements [18]. These shortcomings, such as the omission of deformation from self-weight and missing physical properties, create potential risks for the safety and reliability of entire projects [19].

In this paper, we propose an integrated approach to undertake the aforementioned assembly problems via digital twin technology, which combines 3D reconstruction, position tracking, physical simulations, and dynamic pose control to achieve full-loop tracking and control of the assembly and manufacturing process. The rest of the paper is organized as follows. In Section 2, we review the technologies of structural steel assembly, 3D reconstruction, pose tracking, and dynamic pose control to identify research gaps. In Section 3, we present the methodology and framework of the digital-twin-based high-precision assembly of a steel bridge tower. In Section 4, we analyze and discuss a case study. Finally, in Section 5, we draw conclusions, followed by a discussion of future research directions.

## 2. Literature Review

### 2.1. Steel Assembly

Steel assembly includes the process of lifting, placing, aligning, and assembling steel sections and components to ensure the implementation of construction [20]. A typical method of structural steel installation is to implement a schedule and a detailed plan. The content of this plan includes (1) access to the work, (2) the type and capacity of the proposed equipment, (3) a sequence of operations, (4) detailed crane positions and locations, (5) loads and positions, (6) details of temporary falsework, (7) a method to provide temporary support for stability, (8) details of lifting, and (9) completion [21]. The assembly process, however, has uncertain conditions and risks due to changed site conditions, schedule delays, fabrication mistakes, etc. [22]. More time, higher costs, and further logistics for this procedure bring many challenges, so a worthwhile alternative paradigm, such as a digital-twin-based solution, is greatly needed [23,24].

### 2.2. 3D Reconstruction

Three-dimensional reconstruction indicates the establishment of an object with a 3D shape with computer representations and digital models [13]. Two principal methods are used for 3D reconstruction: triangulation (measuring the distance to a target by measuring the angle of a target point with respect to a known endpoint of a fixed reference line) and time-of-flight (TOF) measurement (using the flight time between two asynchronous

transceivers to measure the distance between nodes) [25,26]. Regarding applications in civil engineering, 3D reconstruction techniques are categorized by point-cloud-based models, mesh-based models, and geometric-based models [25,27]. The 3D reconstruction procedure typically includes three parts: (1) 3D data acquisition, (2) data processing and assembly, and (3) segmentation and matching (also known as modeling) [28–30]. To address the most common technologies that are used in 3D data acquisition devices, a summary (Table 1) is shown below.

**Table 1.** Common technologies and devices used in 3D reconstruction.

| Research Interest | Data Acquisition | Working Principle | Author |
|---|---|---|---|
| As-built BIM | Laser scanning | TOF | Tang et al. [29] |
| As-is BIM | Imaged-based techniques | Triangulation | Lu and Lee [31] |
| Cost, specifications, and applications | Laser scanning | Triangulation and TOF | Parn and Edwards [32] |
| Applications | Photos, videos and laser scanning | Triangulation and TOF | Son et al. [33] |
| Tracking and detection | Computer vision-based | Triangulation | Teizer [34] |
| Defect detection and condition assessment | Computer vision-based | Triangulation | Koch et al. [35] |

In the steel assembly process, the typical application of 3D reconstruction techniques involves quality control and assessment [32]. Automated data acquisition, as a support for quality control and assessment, requires quick methods to collect geometry information, such as that obtained with laser scanning and photogrammetry [36]. Zeibak-Shini et al. stated that final data can be validated by laser scanning and post-inspection [36]. Wang et al. [37] and Monserat and Crosetto [38] explained that Terrestrial Laser Scanning (TLS) devices or combinations with photogrammetry have advantages for structure deformation measurements, especially geometric irregularities, by cross-validating the data with as-designed BIM data.

*2.3. Position Tracking*

Positioning means generating the location of an object, which is referred to as a point that is typically related to the origin of a given coordinate system [39]. Various precision positioning activities help reduce errors in engineering projects [40]. The percentage of sensors used for precision positioning is, i.e., 75% (optical encoder), 12% (laser interferometer), and 6% (capacitive sensor) [39]. To address the most common technologies that are used in position tracking, a summary (Table 2) is shown below.

**Table 2.** Common measuring systems and details of position tracking.

| Measuring System | Measuring Range | Maximum Permissible Error (MPE)/Measurement Uncertainty |
|---|---|---|
| Laser tracker [41] | 150 m | Distance: $\pm 10\ \mu m$<br>Angle: $\pm(15\ \mu m + 6\ \mu m/m)$ |
| Laser tracer [41] | 20 m | $\pm 3\ \mu m/m$ |
| Total station [41] | 3 km | Distance: $\pm(0.6\ mm + 1\ ppm)$<br>Angle: 0.5 *arc second* |
| Laser range scanner [41] | 300 m | Distance: $\pm(1.2\ mm + 10\ ppm)$<br>Angle: 8 *arc second* |
| Leveling [42] | 50~200 m | Millimeter-level |
| Theodolite [43] | 1 km | 0.02 *mm* |
| Close range photogrammetry [44] | 50 m | $5\ \mu m/m$ |
| UAV [45] | Camera-based | Centimeter-level |

One of the most economical ways to track progress automatically is by recording video or taking images, as positioning tolerance for laser trackers demands frequent calibration [34]. Diekman et al., for example, used manual video recording and interpretation to successfully demonstrate steel assembly at high elevations [46]. However, high-precision steel assembly requires a higher standard for precise, reliable, and accurate procedures, which means that the data collected by the aforementioned measuring systems need to be regarded as part of a data repository [47]. Digitization is linked to modern civil engineering design procedures [48]. Unlike previous methods of physical assembly, digital-twin-based pre-assembly with the Building Information Modeling (BIM) platform is capable of simulating complex structures throughout the whole lifecycle from the production line, without any composing elements or components transported out of a factory [23,47]. Position tracking technology coupled with digital-twin-based technology makes the whole construction process effective in terms of time, cost, and high-precision control.

### 2.4. Dynamic Pose Control

Dynamic pose control regards construction performance in automation. To ensure better construction performance, an effective method for pose control is to synchronize a link between the in situ space and the simulation space [49]. BIM has been used widely around the world in the civil engineering field. The combination of BIM and finite element analysis (FEA), also known as the BIM-to-FEA model, has been applied to optimize the assembly process in simulations [50]. For components that are manufactured by numerical control machines, routine checks are necessary [23]. Another typical method for precision control is matching every surveyed point with both the as-designed model and the surveyed coordinates. Each splice plate must process the same operations to ensure that the topological relationship of the points refers to the right nodes. Joining a node and the components of a structure during the steel assembly process concerns positioning, sequencing, safety, and welding strain–stress control [23,51].

### 2.5. Research Gaps

In summary, conventional steel assembly still has room to improve in terms of time, cost, and logistics. Three-dimensional reconstruction technology helps to upgrade the steel assembly process, but it is limited by the precision of data acquisition devices. Currently, the permissible limit of error for steel assembly is controlled by manufacturing deviations and assembly variations, such as vertical inclinations and horizontal deviations. The previous studies focused on the data acquisition process of a component's axis line, containing the manufacturing offset and re-marking the cutting of the component after the axis line is determined [52–54]. Furthermore, this separation of data acquisition, manufacturing, and assembly limits the performance of steel structures and leads to inefficient operation in project management.

## 3. Methodology

In this study, we propose the use of a three-point positioning technique to monitor deformation caused by uncertainty. Generally, deformation types consist of those of gravity and geometry, which are normally caused by the processes of assembly and manufacturing, respectively. Because of these factors, the proposed method uses a loop to handle manufacturing deviations and assembly variations. The whole procedure of the proposed method (see Figure 1 below) includes the topside workflow, indicating the solution to manufacturing deviations, and the downside workflow, indicating the solution to assembly variations.

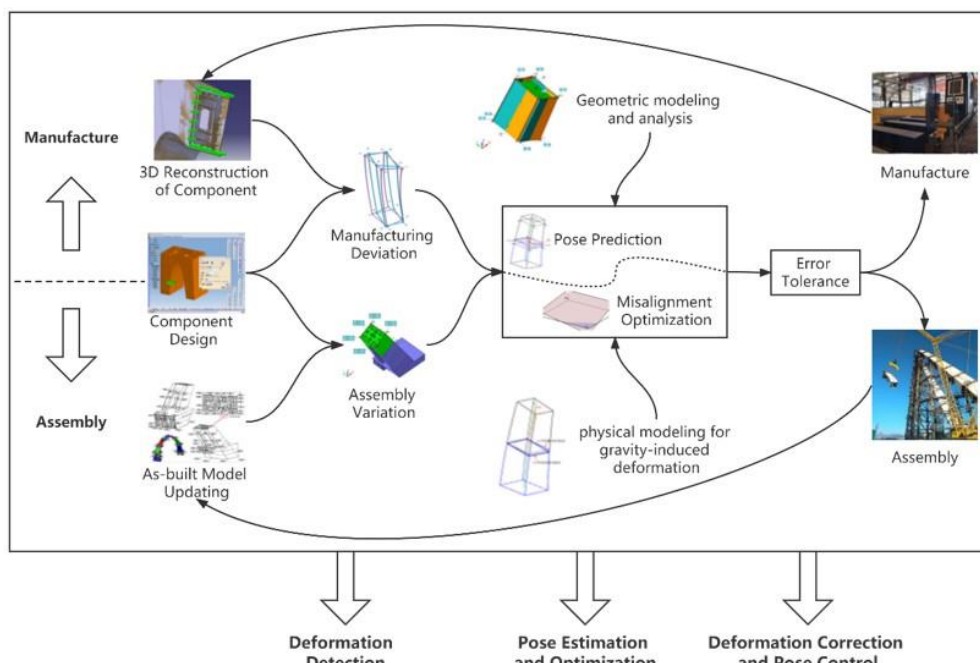

**Figure 1.** Flow chart of digital-twin-based precision assembly.

The entire procedure consists of three main parts: (1) deformation detection, (2) pose estimation and optimization, and (3) deformation correction and pose control. First, under the module of deformation detection, the as-designed model is created at a factory. This model starts in the design phase and contains detailed data for each component of the entire project. After the as-designed component model is produced, the three-point positioning stage can begin. The three-point positioning technique can be used to determine the deformation of each component, involving two submodules: (1) checking for manufacturing deviations and (2) checking for assembly variations. For the former submodule, components from the as-designed model are double-checked to validate their feature point data information after being manufactured and sent from the factory. The other submodule, checking for assembly variations, measures the error that must be considered for all assembled components after the assembly procedure. Second, the deformation information is sent to the next module, pose estimation and optimization. This module involves influences that are caused by gravity and geometry. This integrated module handles pose estimation and misalignment optimization. Regarding pose estimation, the influence of geometry is predicted, and the result indicates the permissible limit of error tolerance for the module of deformation detection. If the requirements are satisfied, then the component undergoes the assembly procedure in situ; otherwise, the dummy component is returned to the as-is model and is adjusted, going through the procedure again. Moreover, under the pose estimation and optimization module, the misalignment optimization submodule receives the information from the last module and sends the data to the computing stage. The result determines the deformation after accounting for the influence of gravity influence, and then it sends the result to determine whether it meets the error tolerance. If it does not, then the dummy component is returned to the as-built model, gets upgraded, and continues through the whole procedure. Once the result satisfies the permissible limit, the component is assembled with fine-tuning, and this part of the as-built model is updated.

### 3.1. Deformation Detection

Information regarding the deformation caused by manufacturing deviations and assembly variations can be determined by monitoring and tracking. The pose of component *i* utilizes the expected state before being manufactured, and then the pose is corrected with 3D reconstruction techniques, producing an updated model for the next step. Using a

combination of BIM technology and 3D reconstruction techniques, a point cloud model can be generated by scanning structures with laser scanners, and then the deformation can be cross-validated between the digital-twin-based model and the as-is model to analyze the manufacturing deviation (shown in Figure 2 below). Three-dimensional scanning is mainly used for the analysis and evaluation of linearity and the dimensions of the space curve plate and the space curve components of the steel tower. With data acquisition from the components of this complex structural steel tower, professional software can process the data and can then generate a digital twin model. We used the least squares method to compare and analyze the linear deviation of the curved plates. Lastly, a deviation report of each component is released once a contrastive analysis is conducted. However, the pose of component *i-1* needs to account for the self-weight influence of component *i* after components *i-1* and *i-2* are assembled, and then the pose of the as-built model can be updated.

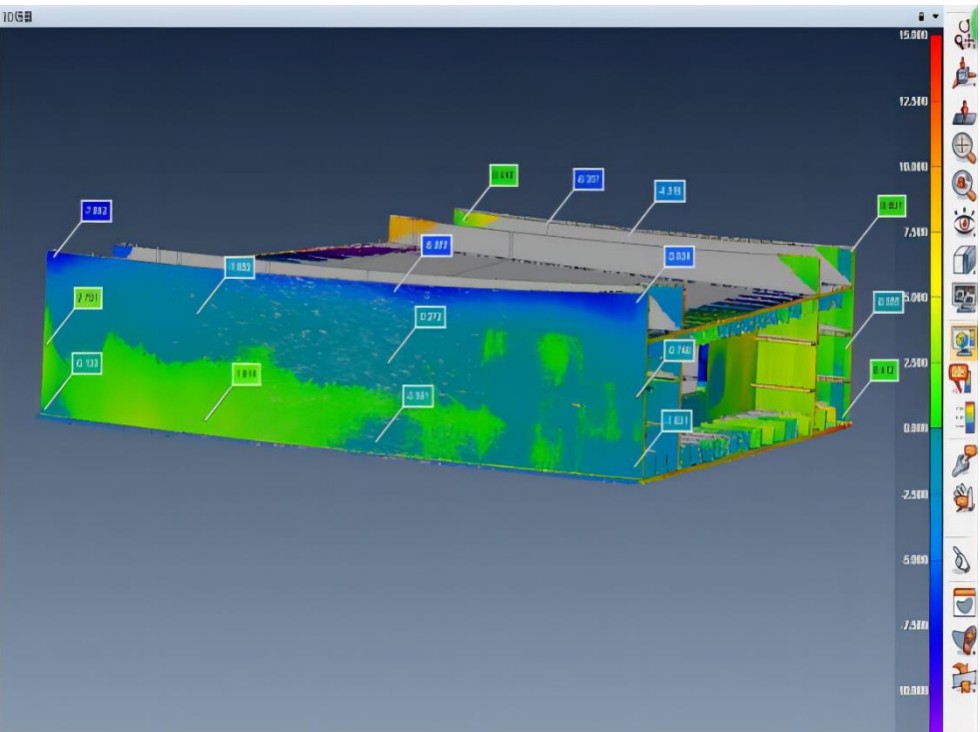

**Figure 2.** BIM-based detection.

Data from data acquisition are then processed for component *i-1*, which must be assembled in situ, and for the as-built model of component *i*. Then, component *i* is used to determine the optimal pose by considering manufacturing deviations, and component *i-1* is docked with the plane of the connection port.

*3.2. Pose Estimation and Optimization*

After data acquisition, deformations are detected and analyzed and are then pushed to the pose estimation and optimization module. When component *i* is set up, the port shape changes due to the function of the dead weight of component *i*, as shown by the red plane and feature points $J_i^m$ in Figure 3. Its non-deforming state needs to account for the correction of the dead weight of component *i*, and the correction amount is calculated according to Formula (1).

$$H_{ei-1,i} = H_{ei-1,\ i-1} + \Delta_{i-1,\ i} \tag{1}$$

$H_{ei-1,i}$ is the form of the upper port of the component without deformation after the assembly of component *i*, $H_{ei-1,\ i-1}$ is the form of the upper port of component *i-1* without

deformation after the assembly of component *i-1*, and $\Delta_{i-1,\,i}$ is the modification of the port shape of component *i* due to the function of the component's self-weight.

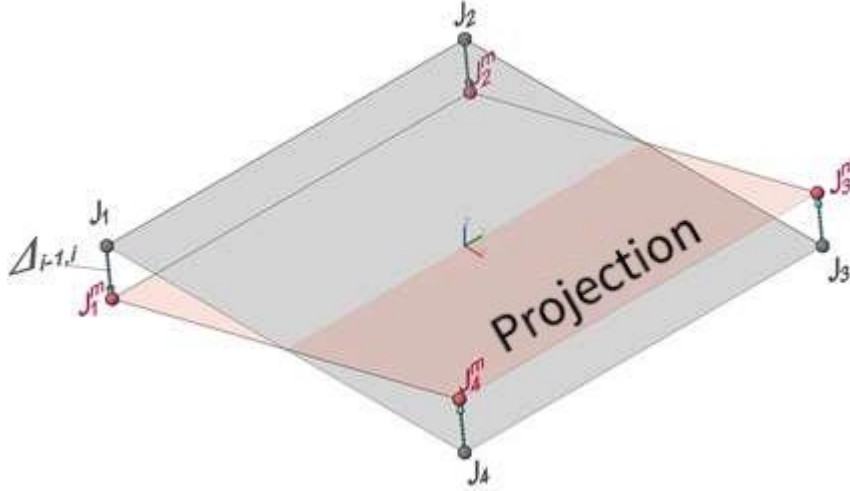

**Figure 3.** Port shape changes.

Figure 4 indicates the misalignment of the side panel. Component *i*, which must be assembled, and the already assembled component *i-1* are connected by docking. Therefore, the pose prediction of component *i* under the premise that component *i-1* has a deviation can be performed by finding a pose with the minimum value of misalignment at the connecting port. This stance is the optimal pose that is assembled in situ.

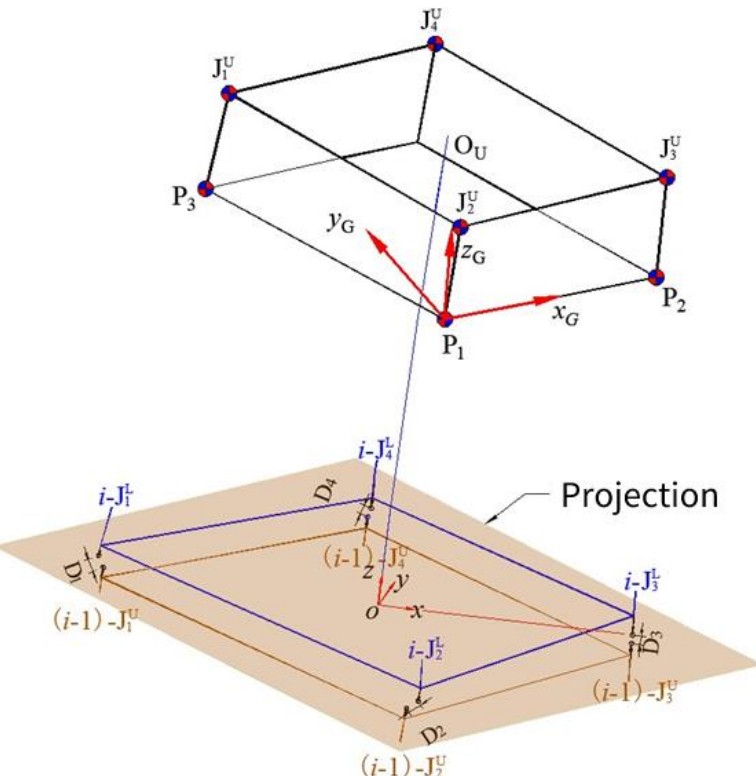

**Figure 4.** Alignment variation.

In the process of determining misalignment optimization, the assembled component *i-1* remains still, and component *i* can only be translated along the x- and y-axes and rotated about the z-axis in a local coordinate system.

In order to account for the optimal assembling pose of the component with a manufacturing deviation, the port docking condition of the manufacturing geometry needs to be analyzed after the welding process. First, the hammer line is set at the upper port, which, based on the setup of the reaction adjustment system at the lower port, is applied to the steel tower containing variable cross-section features and irregular geometry. Adjusting the component vertically when the component's length is exceeded is difficult. Due to this limitation, the optimal pose method based on the 6D (three-way translational displacement and three-axis rotation angle) parameter space is applied to the error analysis and precision control system for manufacturing the components of the large-scale steel tower. This system mainly consists of a force adjustment system, a precision tracking measurement system, and a 6D parameter space attitude optimization computation module.

The schematic diagram of fabrication deviation is shown in Figure 5, where yellow represents the manufacturing state with deviations, and the dark green represents the set state of the expected effect.

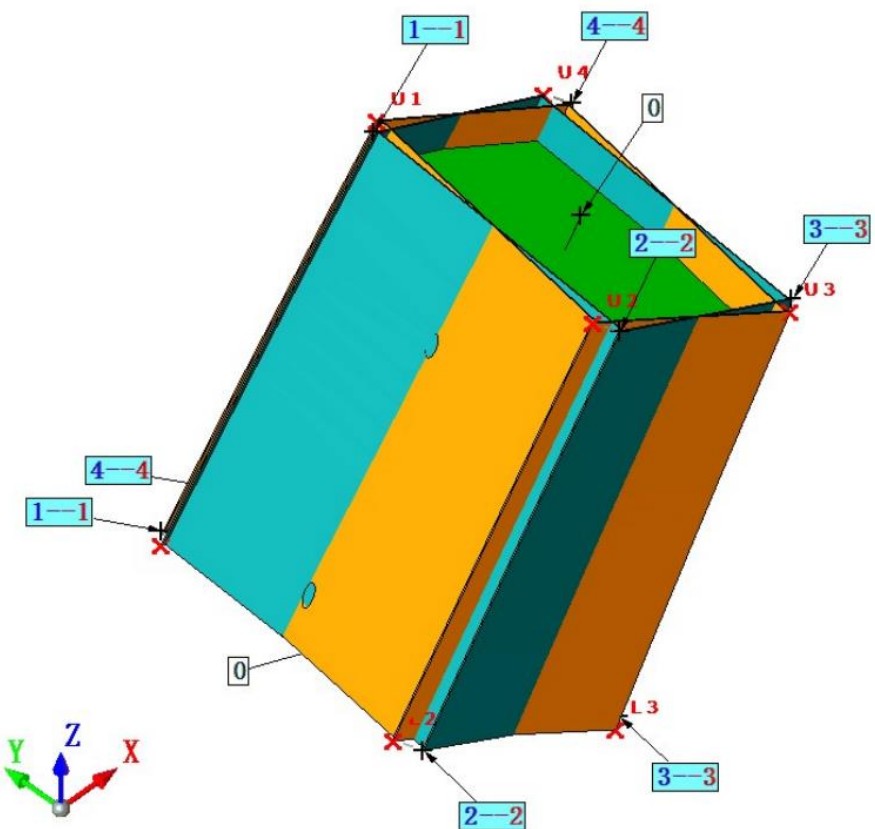

**Figure 5.** Fabrication deviation.

The computation procedure of the prediction model is detailed as follows:

Spatial pose optimization needs to satisfy the principle of the minimum error of control points. See Formulas (2) and (3) for constructing the objective function.

$$f(d_i) = \sum_{i=1}^{4} d_i^U + \sum_{i=1}^{4} d_i^L \tag{2}$$

$$d_i^2 = (X_{ti} - X_{ei})^2 + (Y_{ti} - Y_{ei})^2 + (Z_{ti} - Z_{ei})^2 \tag{3}$$

$d_i$ is the distance between the expected state of each geometric feature point without manufacturing deviations and the assembly state with manufacturing deviations, $U$ is the upper port, $L$ is the lower port, $X\backslash Y\backslash Z$ are the coordinates of feature points, t is the expected state without manufacturing deviations, and e is the assembly state with manufacturing deviations.



The geometric state of site assembly should account for the influence of the dead weight of the new component, and the calculation formula is as follows:

$$H_{ei,1} = H_{ei,0} + D_{tij,j=i} \tag{4}$$

The value of $f(d_i) = \sum_{i=1}^{4} d_i^U + \sum_{i=1}^{4} d_i^L$ can be minimized by solving translation and rotation parameters. The mathematical model is shown in Formula (5).

$$minf(\Delta X, \Delta Y, \Delta Z, \omega_X, \omega_Y, \omega_Z) = \sum \left( (X_{ti} - X_{ei})^2 + (Y_{ti} - Y_{ei})^2 + (Z_{ti} - Z_{ei})^2 \right)^{\frac{1}{2}} \tag{5}$$

See Formula (6) for the coordinate calculation method of inversion from a local coordinate system to a geodetic coordinate system.

$$P^{(1)} = R_l^{-1} \left( R_c \left( R_l \left( P^0 + T_l \right) + T_c \right) - T_l \right) \tag{6}$$

The coordinate transformation calculation method for rigid body translation or for the rotation of an object in the same coordinate system is shown in Formula (7).

$$P_{n \times m}^{e1} = R_{n \times n} P_{n \times m}^{(e0)} + T_{n \times 1} \tag{7}$$

$P_{n \times m}^{(e0)}$ is the coordinate matrix before transformation, $P_{n \times m}^{e1}$ is the coordinate matrix after transformation, $R_{n \times n}$ is the space rotation matrix, $T_{n \times 1}$ is the coordinate translation vector, $n$ is the dimensionality (in this study area, $n = 3$), $m$ is the number of feature points (in this study area, $m = 13$), and $R_{n \times n}$ is related to the rotation of the target about the $X \backslash Y \backslash Z$-axes, as seen in Formulas (8)–(11).

$$R(\omega) = R(\omega_Z) \cdot R(\omega_Y) \cdot R(\omega_X) \tag{8}$$

$$R(\omega_X) = \begin{bmatrix} 1 & 0 & 0 \\ 0 & cos\omega_X & sin\omega_X \\ 0 & -sin\omega_X & cos\omega_X \end{bmatrix} \tag{9}$$

$$R(\omega_Y) = \begin{bmatrix} cos\omega_Y & 0 & -sin\omega_Y \\ 0 & 1 & 0 \\ sin\omega_Y & 0 & cos\omega_Y \end{bmatrix} \tag{10}$$

$$R(\omega_Z) = \begin{bmatrix} cos\omega_Z & sin\omega_Z & 0 \\ -sin\omega_Z & cos\omega_Z & 0 \\ 0 & 0 & 1 \end{bmatrix} \tag{11}$$

$T$ is related to the translational displacement of the target's relative coordinate axes ($X$, $Y$, and $Z$), as seen in Formula (12).

$$T = \begin{bmatrix} \Delta X & \Delta Y & \Delta Z \end{bmatrix}^T \tag{12}$$

To optimize the algorithm, a fast descent method is introduced, such as the Newton method, the conjugate gradient method, and the variable metric method, which all belong to several effective optimization algorithms derivative of the descent method [55]. The fastest descent method is an algorithm obtained from the first approximation of the function, converging quickly in the first few steps but becoming slower and slower [56]. The Newton method, the conjugate gradient method, and the variable metric method are based on the quadratic approximation of a function and have fast convergence speeds. For small- and medium-scale unconstrained optimization problems, variable metric methods, especially BFGS, are effective [55,56].

### 3.3. Deformation Correction and Pose Control

Next, the module of deformation correction and pose control proceeds to adjust the upcoming component according to data acquisition from the last module, and it keeps correcting and controlling the precision of the component. Analysis of the state of the docking port can validate whether it is within the error tolerance and can estimate the pose for component *i* after assembly. Component *i* is determined to decide whether it needs preprocessing, and assembly with geometry positioning is conducted according to the feedback. If the feedback is within the error tolerance, then geometry positioning starts after welding in situ, and component *i+1* undergoes pre-work. If not, then it proceeds with fine-tuning.

As shown in Figure 6a, the green part in the figure is the expected position of the axis line, and the red part is the position of the axis line with deformations. The main reason that it affects the horizontal deformation of the axis is that the angle of the end face between the assembly component and the component to be assembled deviates from the expected state. On the one hand, this could cause the accumulation of variations in the assembly components; however, this problem could be a result of the manufacturing quality of the components. Therefore, a feasible solution is to adjust the elevation of the corners of the lower port of the component to be assembled, i.e., plug gaskets of different thicknesses to compensate for the offset of the end face. As shown in Figure 6b, the goal is to reduce or correct the offset of the axis of the component that is assembled. This method can be used in the linear control of bolted or welded steel towers and welded main girders. For regular vertical steel towers, the thickness of each corner plug pad can be calculated with the ratio of the length to the width of each lower port.

$$\overrightarrow{n_{i0}} = \left(x_{ji}^1, y_{ji}^1, z_{ji}^1\right) - \left(x_{ji}^0, y_{ji}^0, z_{ji}^0\right) \approx z_{ji}^1 - z_{ji}^0 \tag{13}$$

where $\overrightarrow{n_{i0}}$ is the vector of the gasket thickness, $\left(x_{ji}^1, y_{ji}^1, z_{ji}^1\right)$ are the coordinates of four feature points of the upper port that is assembled after transformation, and $\left(x_{ji}^0, y_{ji}^0, z_{ji}^0\right)$ are the coordinates of four feature points of the upper port that is assembled before transformation.

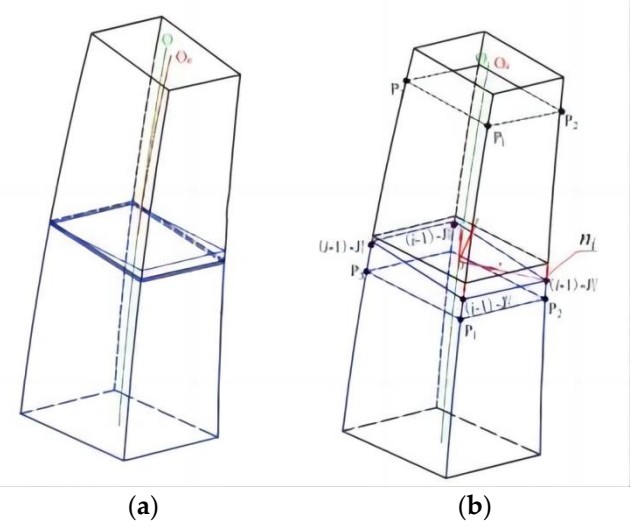

(**a**)          (**b**)

**Figure 6.** Axis line offset and adjustment. (**a**) Axis line offset. (**b**) Adjustment.

In this case, the steel bridge tower is a non-uniform-space structure, and its port has a tilted state. Therefore, precisely solving the gasket thickness of the geometry scale is difficult. As a result, combining the properties of the bridge with the analysis of assembly component variations on-site yields a mathematical algorithm (see Formula (13)), which is suitable for the calculation of the gasket thickness under four corners of the lower port to

adjust the non-uniform steel tower's position. This is convenient for the precision control of the assembly variations in the cable tower, as shown in Figure 7.

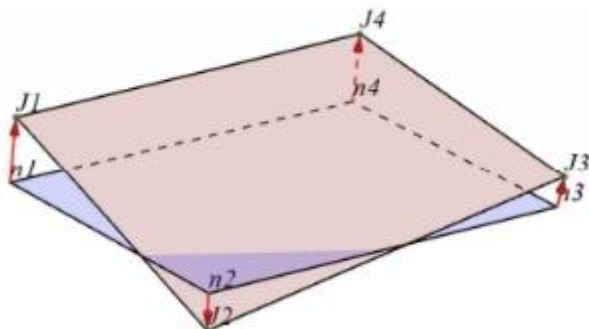

**Figure 7.** Direction of elevation adjustment of corners.

The ABAQUS model can accurately calculate the displacement resulting from self-weight in each or any particular construction stage (as shown in Figure 8 below). The pre-deformation value is calculated in ABAQUS according to the load of the dead weight. In the simulation, the pre-adjustment value of construction and the action of the self-weight load are considered to ensure that the displacement of the model, under the action of the self-weight load after pre-deformation, is as small as possible.

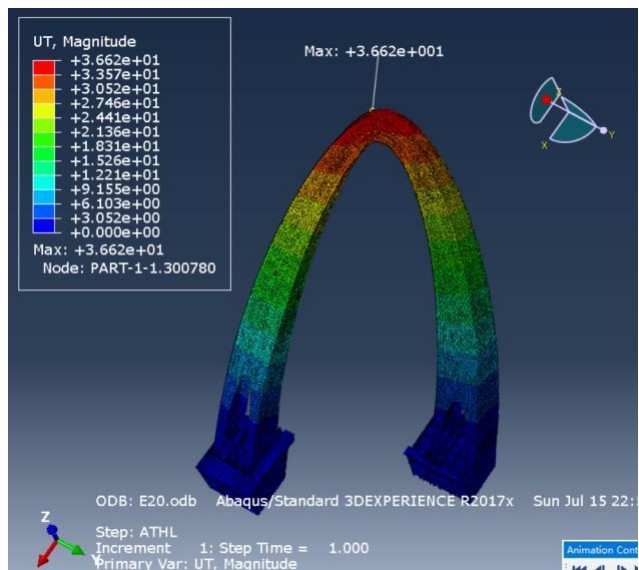

**Figure 8.** Gradually constructed model of the steel tower.

In this study, we incorporate the area of the components of the bridge tower that are installed, which include wall plates, diaphragm plates, and anchor boxes. Furthermore, the computing method for the axis line, which contains deviations in manufacturing, includes setting the hammer line on the upper port based on the setting force of the reaction adjustment system of the lower port, and the optimization method of the space attitude is based on 6D parameters, referring to the three-way translational displacement and three-axis rotation angle.

In general, the pose reference system based on the three-point positioning technique requires a layout of three measurement points, $P_1$~$P_3$, which are located adjacent to the upper port. Then, the temporary coordinate system that was constructed during the as-built coordinate acquisition procedure is organized as $(OXYZ)_F$. This local coordinate system, $(OXYZ)_F$, is constructed with three measurement points. It packages all feature points from the upper port of the component to the local coordinate system that was constructed with those three measurement points. The origin of this local coordinate system, $(OXYZ)_F$,

is located at $P_1$. The x-axis is set as the direction from $P_1$ to $P_2$. The z-axis is the normal direction of the plane formed by the three measurement points, $P_1 \sim P_3$. The y-axis is established according to the right-hand rule. Thus, the layout of the as-built component coordinates is shown in Figure 9.

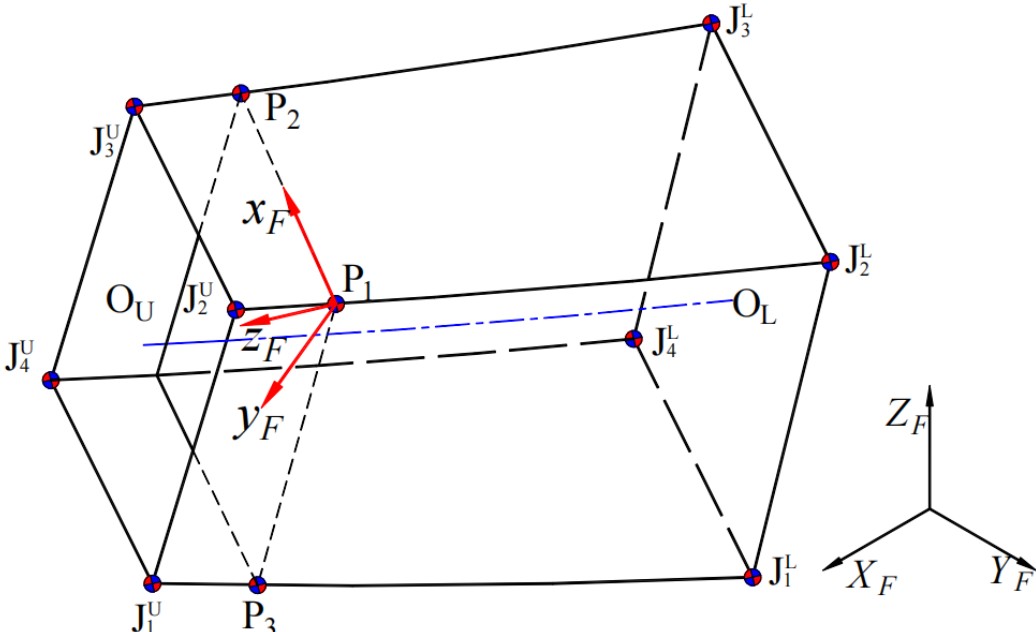

**Figure 9.** Layout of coordination of component *i* at factory.

The geodetic coordinate system is organized as $(OXYZ)_G$. The local coordinate system, $(oxyz)_G$, constructed with the three measurement points based on this geodetic coordinate system, uses the same rule to establish this system. The formula shown below is used to transform the coordinates of any measured feature points from the temporary coordinate system $(OXYZ)_G$ to the local coordinate system $(oxyz)_G$.

$$(x,\, y, z)_G^T = R_G((X,\, Y, Z)_G^T + T_G) \tag{14}$$

A single component has a total of 13 geometry control points, as shown in Figure 10 shown. $J_i^U$ represents the control points of a component of the upper port, and similarly, $J_i^L$ represents the control points of a component of the lower port. Control points, in this case, are located at an intersection directly obtained using the coordinates of the upper wall of a component. $O^U$ is the pivot point at the upper port of a component, and $O^L$ is the pivot point at the lower port of a component. The space between the pivot point of the upper port and lower port is the position of the simulation, and it is not involved in the data acquisition stage. $P_i$ represents the measurement points used to locate the component using the three-point technique. These layout points are somewhat flexible. They are generally set up near the outer side, where the first diaphragm of the upper port intersects with the side panels. The main reason for this placement, according to the relative position of each measurement point, $P_i$, is that it is stable as a result of the rigid constraints of the diaphragm. The distance between the measurement points and the upper port is around 0.5 m~1.5m, and the deformation of the upper port related to the diaphragm is negligible.

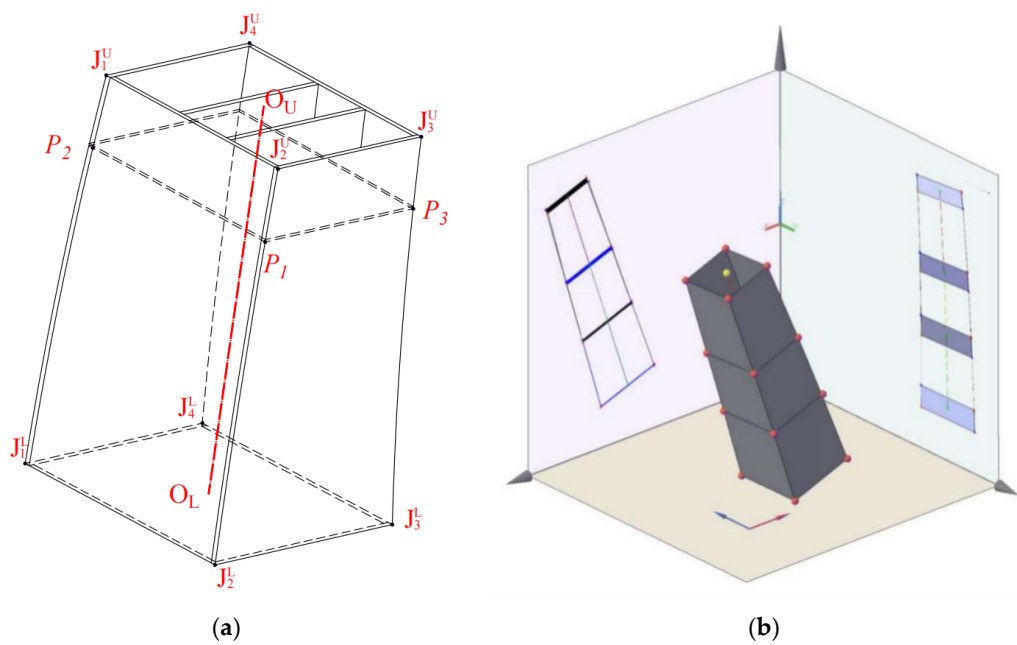

(**a**)                                                                              (**b**)

**Figure 10.** Layout of geometry control feature points. (**a**) Geometry control points of each component. (**b**) Typical pose space comprising three components and the projection in different directions.

According to the computation in Figure 11, port coordinates and deformation correction results for components follow the procedure shown below.

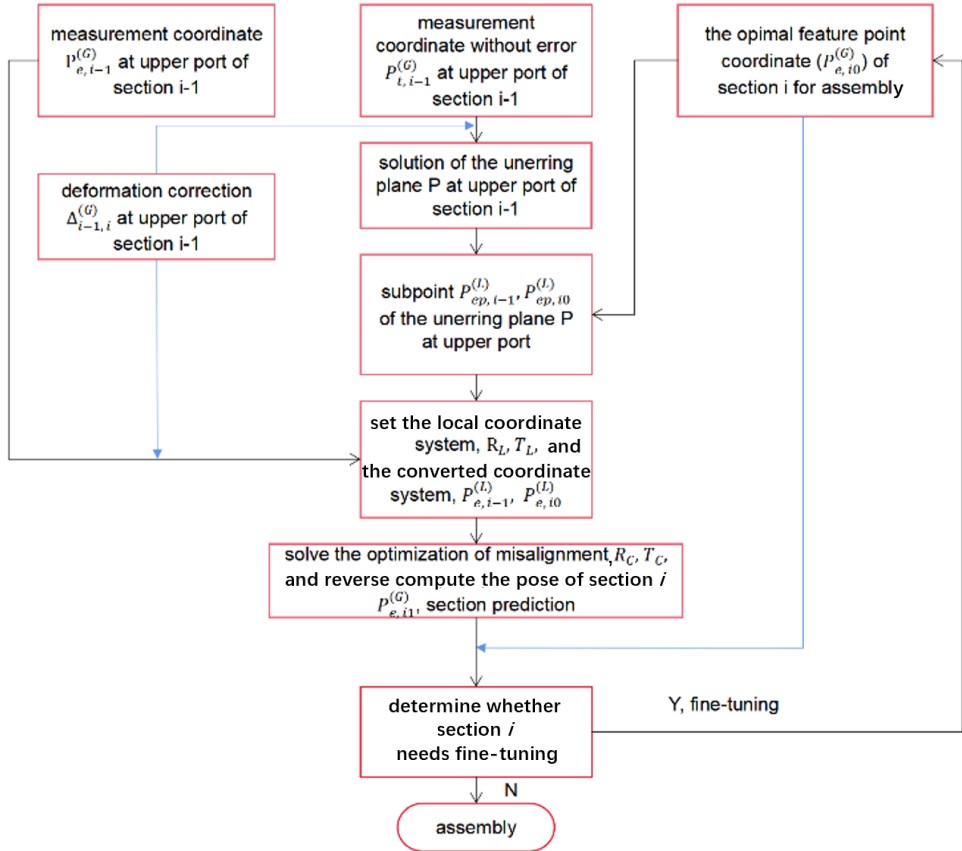

**Figure 11.** Computation of geometry estimation.

## 4. Case Study

The case study (as shown in Figure 12 below)is a steel cable bridge tower with a non-uniform tilt for both limbs, an asymmetric space, and a torsional cross-section fully welded steel box arch. The height of the tower is 124 m. The construction plan for its erection involves lifting segment-by-segment, divided into 31 segments. Unlike the bolted steel structure, the segmental ends of the bridge pylon beams are not machined, and the geometry is subject to manufacturing errors, weld shrinkage, and other factors that greatly affect the bridge. Therefore, with full consideration of the features of the case study, we studied the geometric shape control technology in the manufacturing and assembly stages.

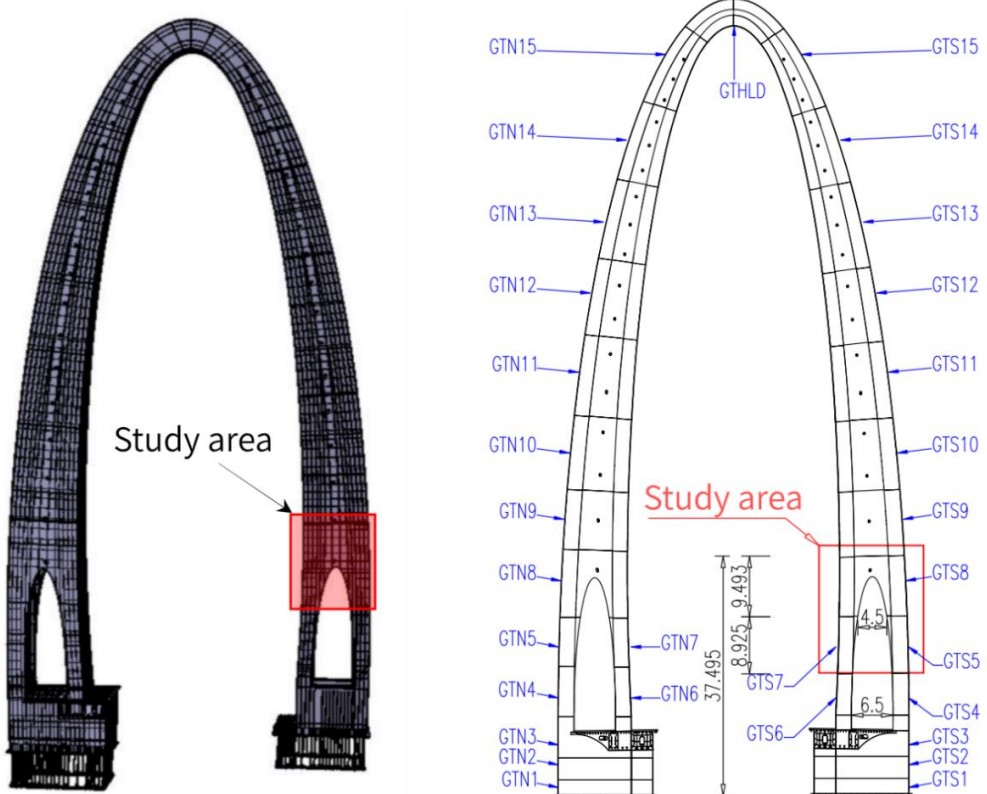

**Figure 12.** Study area.

Component 8 (GTS8) of the southern limb of the high tower of the bridge was selected to predict its assembly state. The position of this component in the high tower is shown in Figure 12. Table 3 shows the coordinates and correction results according to Figure 11.

**Table 3.** Measured data of assembly components and deformation correction (mm).

| No. | Measured Coordinates $P^{(G)}_{e, i-1}$ | | | Deformation Correction $\Delta^{(G)}_{i-1, i}$ | | | Coordinate Correction $P^{(G)}_{em, i-1}$ | | |
|---|---|---|---|---|---|---|---|---|---|
| | X | Y | Z | X | Y | Z | X | Y | Z |
| U1 | 249,471 | 460,458 | 117,987 | −2.7 | −0.1 | −0.9 | 249,468 | 460,458 | 117,986 |
| U2 | 249,449 | 458,423 | 118,009 | −2.7 | −0.1 | −0.9 | 249,446 | 458,423 | 118,008 |
| U3 | 241,540 | 458,513 | 113,679 | −2.1 | −0.1 | −2.0 | 241,538 | 458,513 | 113,677 |
| U4 | 241,568 | 460,547 | 113,656 | −2.1 | −0.1 | −2.0 | 241,566 | 460,547 | 113,654 |
| U5 | 249,365 | 452,146 | 118,072 | −2.7 | −0.1 | −0.9 | 249,362 | 452,146 | 118,094 |
| U6 | 249,347 | 450,111 | 118,095 | −2.7 | −0.1 | −0.9 | 249,344 | 450,111 | 118,094 |
| U7 | 241,467 | 451,996 | 113,748 | −2.1 | −0.2 | −1.9 | 241,465 | 451,996 | 113,746 |
| U8 | 241,455 | 449,963 | 113775 | −2.1 | −0.2 | −2.0 | 241,453 | 449,963 | 113,773 |

Based on Table 4, the plane fitting results for the ports of components GTS5 and GTS7 and the calculation results for the projection coordinates of the corresponding points connected to the ports of the upper component are shown in Table 5. The results show that the port on this component has no deformation, and the maximum distance between each feature point and the corresponding projection point after the target state deformation correction is 0.7 mm. The plane component of the port deformation is assumed to be in good compliance. The deformation is a result of torsional distortion and rounding errors in calculation.

**Table 4.** Expected target state of assembly components and deformation correction.

| No. | Expected Coordinates $P_{t,\,i-1}^{(G)}$ | | | Deformation Correction $\Delta_{i-1,\,i}^{(G)}$ | | | Coordinate Correction $P_{tm,\,i-1}^{(G)}$ | | |
|---|---|---|---|---|---|---|---|---|---|
| | X | Y | Z | X | Y | Z | X | Y | Z |
| U1 | 249,482 | 460,454 | 117,969 | −2.7 | −0.1 | −0.9 | 249,480 | 460,454 | 117,968 |
| U2 | 249,459 | 458,420 | 117,991 | −2.7 | −0.1 | −0.9 | 249,456 | 458,420 | 117,990 |
| U3 | 241,557 | 458,511 | 113,664 | −2.1 | −0.1 | −2.0 | 241,555 | 458,511 | 113,662 |
| U4 | 241,579 | 460,545 | 113,641 | −2.1 | −0.1 | −2.0 | 241,577 | 460,545 | 113,639 |
| U5 | 249,389 | 452,139 | 118,062 | −2.7 | −0.1 | −0.9 | 249,387 | 452,139 | 118,061 |
| U6 | 249,366 | 450,105 | 118,084 | −2.7 | −0.1 | −0.9 | 249,364 | 450,405 | 118,083 |
| U7 | 241,489 | 452,006 | 113,740 | −2.1 | −0.2 | −1.9 | 241,487 | 452,006 | 113,738 |
| U8 | 241,467 | 449,972 | 113,763 | −2.1 | −0.2 | −2.0 | 241,465 | 449,972 | 113,761 |

**Table 5.** XOY plane fitting of local coordinates, and analysis of error.

| No. | Coordinate Correction $P_{tm,\,i-1}^{(G)}$ | | | Coordinate Correction $P_{tmp,\,i-1}^{(G)}$ | | | Distance |
|---|---|---|---|---|---|---|---|
| | X | Y | Z | X | Y | Z | |
| U1 | 249,480 | 460,454 | 117,968 | 249,480 | 460,454 | 117,968 | 0.4 |
| U2 | 249,456 | 458,420 | 117,990 | 249,456 | 458,420 | 117,990 | 0.7 |
| U3 | 241,555 | 458,511 | 113,662 | 241,555 | 458,511 | 113,662 | 0.6 |
| U4 | 241,577 | 460,545 | 113,639 | 241,577 | 460,545 | 113,639 | 0.3 |
| U5 | 249,387 | 452,139 | 118,061 | 249,387 | 452,139 | 118,061 | 0.6 |
| U6 | 249,364 | 450,105 | 118,083 | 249,364 | 450,105 | 118,083 | 0.2 |
| U7 | 241,487 | 452,006 | 113,738 | 241,487 | 452,006 | 113,738 | 0.6 |
| U8 | 241,465 | 449,972 | 113,761 | 241,465 | 449,972 | 113,761 | 0.4 |

Building the objective function with Formula (5) and optimizing the calculations based on the BFGS quasi-Newton method are key to minimizing the value of the error. The initial value is zero in the optimization calculations and is assumed to be the initial state as well as the component assembled in the optimal space position. The value of the objective function with iterative calculations to improve the process is shown in Figure 13. The result shows that the target function value returns a downward tendency. The initial value decreased from 83 mm to 66 mm after 22 iterative calculations. According to the results shown in Table 6, the maximum deformation of each feature point on the upper and lower port projection is 10 mm, and the average value is 5 mm. Although the optimization process does improve the docking error variable, a gap still exists regarding the design requirements, which demand 2 mm as the docking error variable. The above error is a result of manufacturing deviations in the plates and welding shrinkage deformation of the plates. In fact, before formal welding, the local staggered deformation can be adjusted to the thermal orthosis process to ensure that the conditions of panel butt welding are met. Therefore, the optimized parameters (results seen in Figure 13 below) can be calculated to

solve for the pose of the component assembled under the optimal port docking variation, according to Formula (2).

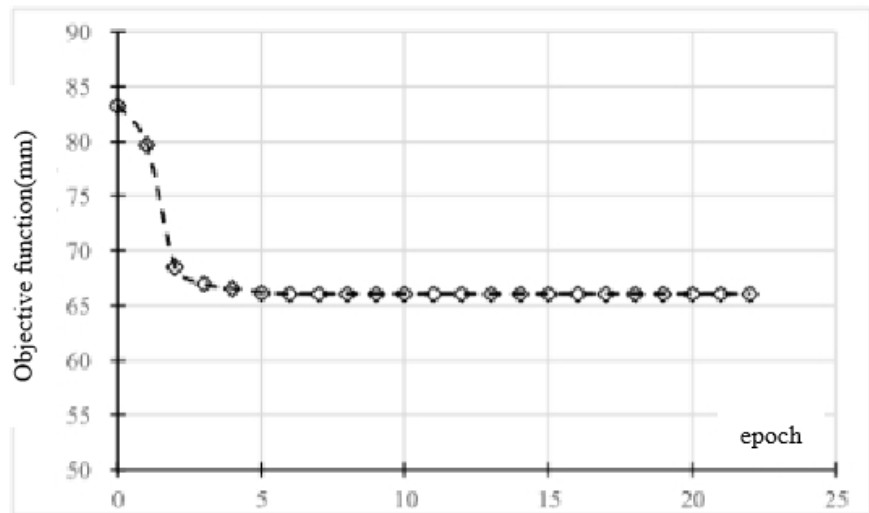

**Figure 13.** Iterative computations of the objective function.

**Table 6.** Optimization state of port docking variation (mm).

| No. | Assembly Components $P_{e,i-1}^{(L)}$ | | | Components to Be Assembled $P_{e,i1}^{(L)}$ | | | Docking Error Variable Calculation | | Normal Vector Overlap |
|---|---|---|---|---|---|---|---|---|---|
| | X | Y | Z | X | Y | Z | X | Y | Z |
| U1\L1 | 10,354 | 2 | 22 | 10,350 | 8 | −19 | −4 | 6 | −41 |
| U2\L2 | 8319 | 1 | 21 | 8316 | 11 | −8 | −2 | 10 | −28 |
| U3\L3 | 8368 | 9018 | 21 | 8368 | 9018 | 2 | 0 | 0 | −19 |
| U4\L4 | 10,403 | 9014 | 18 | 10,400 | 9024 | −11 | −3 | 10 | −30 |
| U5\L5 | 2041 | 17 | 21 | 2023 | 8 | −7 | −7 | −9 | −28 |
| U6\L6 | 6 | 12 | 19 | 3 | 8 | 0 | −2 | −4 | −19 |
| U7\L7 | 1850 | 9020 | 18 | 1859 | 9011 | −3 | 9 | −8 | −21 |
| U8\L8 | −183 | 9008 | 16 | −174 | 9011 | −8 | 9 | 3 | −24 |

Figure 14 shows a distribution of the variations in the feature points of the upper port relative to the optimal assembly attitude of component GTS8 and the variations in the feature points of the lower port relative to components of GTS5 and GTS7, assembled with the optimal number of alternate sides for the connected ports. The variations in the feature points of the upper port, relative to the optimal assembly attitude, correspond to Table 7. Likewise, variations in the feature points of the lower port, relative to assembly components GTS5 and GTS7, correspond to Table 6. Variations in the table refer to the local coordinate system. The value of the overlap of the normal vector indicates the value of the difference in the local coordinate system on the z-axis between the upper and lower ports' feature points. A negative value indicates overlap, i.e., the length does not reach the target. Similarly, a positive value indicates a gap, i.e., the lengths of the upper and lower components are shorter than those of the target. If the port of GTS8 can reach the variation state shown in Figure 14a, the overlap area of the upper and lower component connecting ports should be eliminated; otherwise, GTS8 cannot reach the predicted position due to the component's length. Therefore, the normal vector's overlap quantity for the corresponding feature points of the connected ports in the table is the quantity of the adjustment of the component length. Because components GTS5 and GTS7 are set up while GTS8 is still being set up in the factory, cutting is carried out on each corner point of the lower port of

component GTS8. The predicted variation distribution after cutting is shown in Figure 14b. After processing and adjusting GTS8 according to the normal vector overlap quantity provided in the table, the measured variation in the upper end of the site after assembling and positioning is shown in Figure 15. The variation is close to the prediction variation, and the maximum variation of the axis variation is only 4 mm, indicating that the proposed method can effectively guide the site assembly of the component.

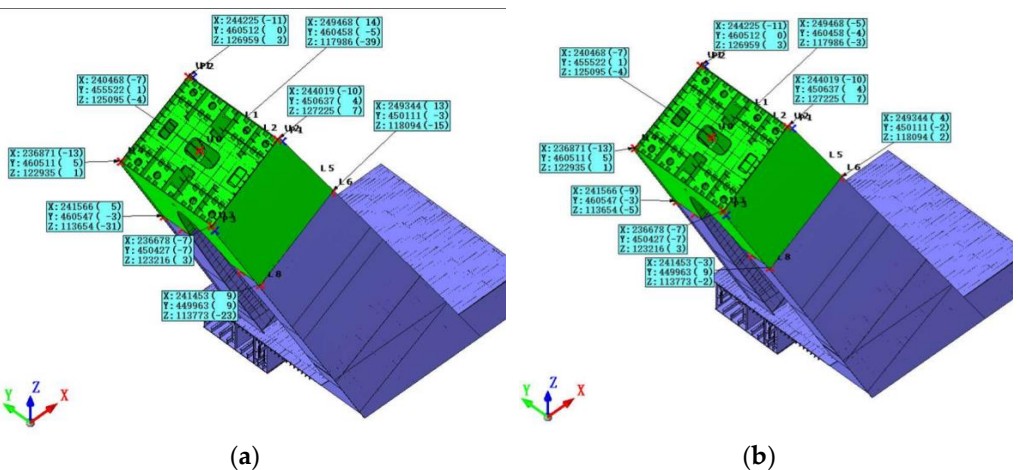

|     (a)     |     (b)     |

**Figure 14.** Prediction of port docking variation for GTS8. (**a**) Component before post-processing. (**b**) Component after post-processing.

**Table 7.** Upper port state of the optimized component to be assembled for docking variation (mm).

| No. | Optimized Coordinates $P_{e,i0}^{(G)}$ | | | Optimized Coordinates on the Lower Port $P_{e,i1}^{(G)}$ | | | Variation | | |
|-----|---------|---------|---------|---------|---------|---------|-----|-----|-----|
|     | X       | Y       | Z       | X       | Y       | Z       | X   | Y   | Z   |
| U1  | 244,223 | 460,509 | 126,967 | 244,214 | 460,512 | 126,962 | −9  | 3   | −5  |
| U2  | 244,014 | 450,638 | 127,235 | 244,009 | 450,641 | 127,232 | −5  | 3   | −3  |
| U3  | 236,676 | 450,420 | 123,222 | 236,671 | 450,420 | 123,219 | −5  | 0   | −3  |
| U4  | 236,867 | 460,516 | 122,941 | 236,858 | 460,516 | 122,936 | −9  | 0   | −5  |
| UO  | 240,468 | 455,522 | 125,095 | 240,461 | 455,523 | 125,091 | −7  | 1   | −4  |
| P2  | 244,688 | 460,482 | 126,557 | 244,679 | 460,485 | 126,552 |     |     |     |
| P1  | 244,473 | 450,565 | 126,815 | 244,468 | 450,568 | 126,812 |     |     |     |
| P3  | 236,916 | 450,301 | 122,679 | 236,911 | 450,301 | 122,676 |     |     |     |

GTS8 is a component used to transition from the limb to the whole component. Regional component stiffness changes significantly, and the stress in the bottom is relatively large. Quality welding connections and assembly forms are strictly required for the component. In this case, the vertical height of the center of the component's end is H = 37.485 m from the tower's root. The error tolerance of the upper port is 9.4 mm (H/4000), and the allowable height difference is ±12 mm (±2 N, n = 6). The thickness of the connecting wall with the preceding component is 46 mm, and the allowable value of the port docking error variable is 2 mm.

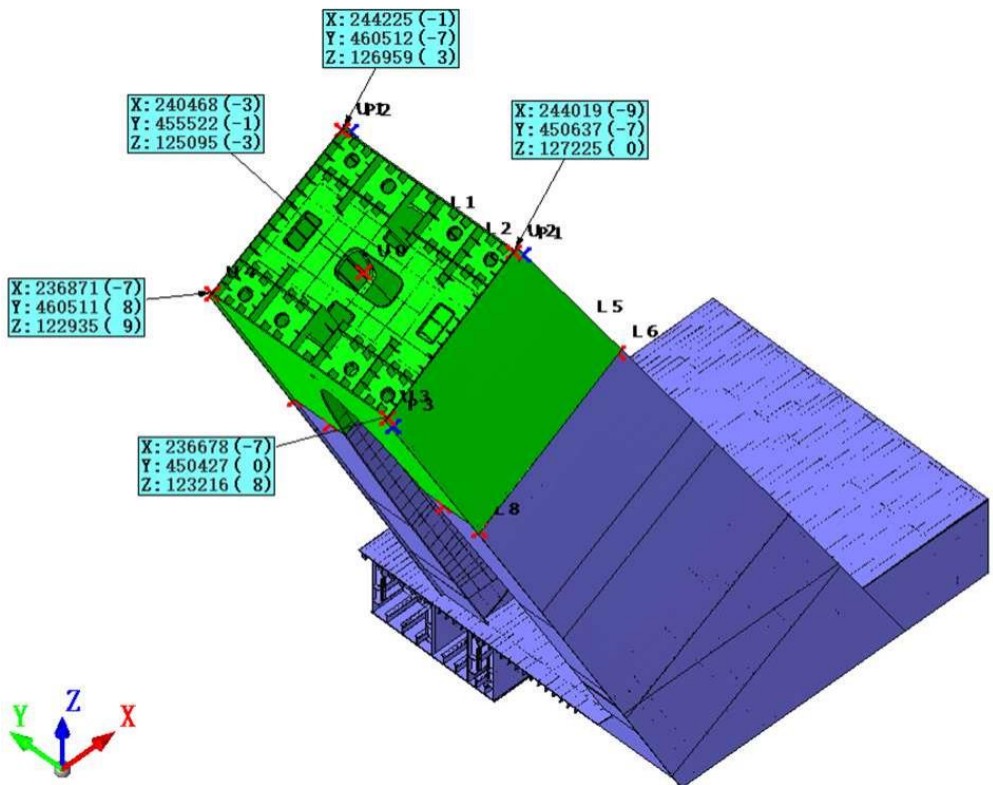

**Figure 15.** Prediction after determining the state of GTS8.

The vertical height of component GTS8 is 9.493 m, and the upper port is a quadrilateral of 8.3 m × 9.9 m. The angle between component GTS8 and the XOY plane space of the geodetic coordinate system is about 28.75°, and the component's weight is about 470 t. The preceding components connected with GTS8 are components GTS5 and GTS7, with a vertical height of 8.925 m. The angle between the plane of the port and the XOY plane space of the geodetic coordinate system is about 28.71°.

In this case study, we applied the three-point positioning technique and completed positioning and measurements. The follow-up inspection and measurements are achievable. Regarding this steel tower, the deviation on the axis of each component for the case study (Figure 12) after bridge closure is shown below in Table 8.

**Table 8.** Axis deviation of steel tower after bridge closure (mm).

| Limbs | Bias | GTN 3 | GTN 4–6 | GTN 5–7 | GTN 8 | GTN 9 | GTN 10 | GTN 11 | GTN 12 | GTN 13 | GTN 14 | GTN 15 |
|---|---|---|---|---|---|---|---|---|---|---|---|---|
| North | $\Delta X$ | 3 | −5 | −6 | −5 | 12 | 4 | −9 | 5 | −18 | 2 | −18 |
|  | $\Delta Y$ | 3 | −3 | −4 | −2 | 3 | −8 | −5 | −4 | −9 | 3 | −4 |
| South | $\Delta X$ | 3 | −8 | −10 | −6 | −12 | −2 | −15 | −18 | −12 | 3 | −12 |
|  | $\Delta Y$ | 2 | 3 | −4 | −10 | −3 | −6 | −7 | −4 | −8 | 4 | −8 |
| Allowance |  | 3 | 5 | 7 | 10 | 12 | 15 | 18 | 21 | 24 | 27 | 30 |

## 5. Conclusions

Gravity causes deformation and non-uniform torsion bending, but digital-twin-based precision control technology for components improves construction ability with respect to geometric pose control. It can also be used to create a technological innovation system with precision control for spatial anomalous structural components. Based on digital twin technology, the model that we present in this study involves a virtual preassembly method with multiple components, combining gravity, a dynamic pose prediction method, and a pose fine-tuning method. First, the three-point positioning technique provides a fast and precise method that can be used to upgrade the coordinates of each component, and these

features can help with the application of virtual preassembly. As the upper and lower workflows proceed, the pose of each assembled component is predicted and optimized dynamically. This precision control technology shows positive results, as follows: (1) the axis variation in the tower is less than H/4000 and goes down to H/6000, and (2) the elevation variation is less than 20 mm.

This method avoids a defect of virtual preassembly technology, which only considers the non-stress state and cannot predict the assembly state of torsion bending in the component on-site. Moreover, this method's pose optimization utilizes the predicted position when the component to be set up is undergoing connection, guiding the way to subsequent component deformation correction and helping to make precise adjustments during the process of assembling a cable tower. In the construction of complex structures, the application of a digital-twin-based model is conducive to the inspection and verification of the structure, allowing for the ease of tracing the causes of problems of quality, and it is conducive to handling problems in a timely manner, ensuring steady progress for quality control and assessment.

Future prospects include adding new factors to the digital-twin-based model, such as wind load, temperature variation, hygral changes, and variations in the process. All these factors can be packaged into the property panel that is embedded in the digital twin model. Moreover, a new optimization algorithm can be used and can replace the current computation procedure. Last, with the development of mapping equipment, obtaining feature points from the data acquisition step can be easier, faster, and more precise, thus potentially pushing the industry innovatively.

**Author Contributions:** Methodology, J.L. and Q.L.; formal analysis, Q.M.; investigation, J.L. and H.X.; writing—original draft preparation, J.L.; writing—review and editing, Q.L., Q.M. and H.X.; project administration, J.L. All authors have read and agreed to the published version of the manuscript.

**Funding:** This research was funded by Beijing Scholars Program.

**Data Availability Statement:** The authors do not have permission to share data.

**Acknowledgments:** This study was supported by the Research Development Project of the Ministry of Housing and Urban–Rural Development of the People's Republic of China (K20210032).

**Conflicts of Interest:** The authors declare that they have no competing interests.

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
