# Peer review of "Digital-Twin-Based High-Precision Assembly of a Steel Bridge Tower"

_buildings, doi:10.3390/buildings13010257_

Round 1
Reviewer 1 Report
The article presents many English language mistakes (i.e. adverbs instead of adjectives: «seriously» instead of “serious”, «well» instead of “good”; errors in the use of verb tenses; errors in the syntactic construction of sentences; etc.) which make some paragraphs of the article extremely difficult to read, and should be deeply proofread by a native English speaker before being evaluated from the point of view of scientific quality.
Reviewer 2 Report
The paper deals with the use of the digital twin as a support for the construction process of steel structures, as for example, bridges.
The topic and the case study proposed by the Authors are very interesting, but at the same time, the work has serious deficiencies in the contents and in their presentation, that must be solved before its publication. The Reviewer comments to improve the paper are reported below:
· In the Introduction, when talking about steel structure benefits, Authors wrote: Furthermore, the earthquake effect can be reduced above 30% ~ 40 %...; this is not well written. Indeed, from a seismic point of view, steel structures are advantageous because of their lower masses, but, at the same time, they are more deformable and consequently they exhibit higher displacements under seismic excitation. Of course, in bridges this issue is faced with the use of proper seismic isolators rather than increasing the stiffness of the structure (sometimes using steel-concrete composite structures). Therefore, Authors are suggested to modify the above sentence to better address this aspect.
· In Section 3.2, Authors proposed many formulae to discuss about the pose estimation & optimization, but they are quite disconnected from the logical thread of the paper. At the beginning, they discuss of the digital twin in the construction process in a general and conceptualized manner, but suddenly they provide information with very high level of detail. Are these formulae a new method proposed by the Authors? If no, the Reviewer suggests to eliminate these equations providing only the fundamental ones and the concepts of their use and advantages.
· The logical scheme of Figure 6 is quite difficult to understand. Again, if it is not strongly necessary, Authors are suggested to eliminate it or, in case they disagree, please add an explanation of its content in the text.
· Overall, all Section 3 has serious concerns. In this section Authors want to provide the general basis of the methodology, but they do not provide a general discussions. In Section 3.2 Authors give a huge amount of details without explain their general purpose and use. The same occurs in Section 3.3; moreover, here, they provide lot of details that are specific for steel towers or girders. So, the Reviewer suggests to re-think the whole Section 3. If the Authors’ goal is that of providing the general basis of the methodology, so please remove the huge amount of details and use a more general discussion, otherwise contextualize the topic for specific cases of application.
· Section 4 is a typical design report, but no references are made about the digital twin and the procedure discussed above. When find a case study, the reader expects to read something about the application of the entire procedure and something about the effectiveness of the digital twin application. None of this is reported in Section 4. Authors are request to discuss about the application of the proposed procedure in the real case study and to reduce the huge amount of technical data about the single structural component of the tower.
Moreover, some minor comments are provided:
· The quality of Figures 2, 8, 13 is very low. Please, improve the figures.
· All tables (included the first two) should be defined with captures and quoted in the text. Please, revise the whole table captures and their citations on text.
Reviewer 3 Report
In this manuscript, author proposes an integrated approach to tackle aforementioned challenges via digital twin technology, which combines three modules: (1) deformation detection, (2) pose estimation & optimization, and (3) deformation correction & pose control, to achieve full-loop tracking and control of the manufacturing and assembling process. The topic is significant and some valuable conclusions are obtained. However, the manuscript is not logically and critically presented in scientific form. It cannot be published in the current form.
1. There are several grammatical errors and format errors in the text that need to be read carefully.
2. 【23】In the abstract, 1/4000 of the 124m equals 4mm?
3. 【46】“120-year”should be “120 years”
4. 【74】In the section 2, " the research gaps "do not explicitly be identified below.
5. 【90】Wrong phrase: ‘lot of’ should be corrected to ‘a lot of ‘ or ‘ lots of’
6. There are some informal English expressions, such as so, very and obviously.
7. 【221】It would be better if author can improve the sharpness of the Figure 2.
8. The author does not clearly describe or express the application process of digital twin in this study and its linkage with the physical model.
9.【95】The two principal methods for 3D reconstruction in the article were not introduced.
10.【231】The number after the formula does not need to be dotted.
11.【281、283】The symbols in the formula are not introduced.
12.【355】Figure 7 is too blurry.
13.【362】The algorithm mentioned in the article was not introduced in detail.
14.【450】Table 1 does not introduce and some symbols in it are vague.
15.【512】The font in the title is incorrect.
Round 2
Reviewer 1 Report
After the linguistic revision the article has greatly improved its readability, however, some unclear or ambiguous passages of the text remain, as well as some inaccuracies.
Please see the attached file for more specific comments.

Reviewer 2 Report
The Authors modified the paper in accordance with the Reviewer's suggestions.
Author Response
Thank you very much for reviewing the manuscript
Reviewer 3 Report
Accept as it is.
Author Response
Thank you very much for reviewing the manuscript.